# Laser Surface Microstructuring of a Bio-Resorbable Polymer to Anchor Stem Cells, Control Adipocyte Morphology, and Promote Osteogenesis

**DOI:** 10.3390/polym10121337

**Published:** 2018-12-03

**Authors:** Rocio Ortiz, Iskander Aurrekoetxea-Rodríguez, Mathias Rommel, Iban Quintana, Maria dM Vivanco, Jose Luis Toca-Herrera

**Affiliations:** 1Ultraprecision Processes Unit, IK4-TEKNIKER, C/Iñaki Goenaga 5, 20600 Eibar, Spain; rocio.ortiz@tekniker.es; 2CIC bioGUNE, Technology Park of Bizkaia, Ed. 801A, 48160 Derio, Spain; iaurrekoetxea@cicbiogune.es (I.A.-R.); mdmvivanco@cicbiogune.es (M.d.V.); 3Fraunhofer Institute for Integrated Systems and Device Technology IISB, Schottkystrasse 10, 91058 Erlangen, Germany; mathias.rommel@iisb.fraunhofer.de; 4Institute for Biophysics, Department of Nanobiotechnology, University of Natural Resources and Life Sciences Vienna (BOKU), Muthgasse 11, 1190 Vienna, Austria; jose.toca-herrera@boku.ac.at

**Keywords:** bio-resorbable polymers, picosecond laser micromachining technology, surface microstructuring, human mesenchymal stem cells, osteogenesis, adipocytes

## Abstract

New strategies in regenerative medicine include the implantation of stem cells cultured in bio-resorbable polymeric scaffolds to restore the tissue function and be absorbed by the body after wound healing. This requires the development of appropriate micro-technologies for manufacturing of functional scaffolds with controlled surface properties to induce a specific cell behavior. The present report focuses on the effect of substrate topography on the behavior of human mesenchymal stem cells (MSCs) before and after co-differentiation into adipocytes and osteoblasts. Picosecond laser micromachining technology (PLM) was applied on poly (L-lactide) (PLLA), to generate different microstructures (microgrooves and microcavities) for investigating cell shape, orientation, and MSCs co-differentiation. Under certain surface topographical conditions, MSCs modify their shape to anchor at specific groove locations. Upon MSCs differentiation, adipocytes respond to changes in substrate height and depth by adapting the intracellular distribution of their lipid vacuoles to the imposed physical constraints. In addition, topography alone seems to produce a modest, but significant, increase of stem cell differentiation to osteoblasts. These findings show that PLM can be applied as a high-efficient technology to directly and precisely manufacture 3D microstructures that guide cell shape, control adipocyte morphology, and induce osteogenesis without the need of specific biochemical functionalization.

## 1. Introduction

Adult stem cells are the main source for developing new strategies in regenerative medicine, such as cell-based therapy [1], genetic therapy [2] and tissue engineering [3]. Proliferation and differentiation of stem cells in vivo are regulated by their microenvironment, known as niche, which comprises both cellular components and interacting signals between them [4,5,6]. These niches, in addition to other functions, provide stem cells with physical anchors (by means of adhesion molecules) and regulate the molecular factors that control cell number and fate [5]. Some of these factors are influenced by cell shape, cytoskeletal tension, and contractility [7,8]. In this regard, the design of biomaterials with architectures that mimic natural cell microenvironments might be a powerful tool to better understand and manipulate cell function as a strategy for future cell-based therapies. Among the biomaterial properties that affect cell behavior, surface topography has shown an enormous potential to control cell shape and location [9]. Several researchers have observed that microscale and nanoscale topographies in the form of pillars, grooves, pits, or pores can induce the differentiation of human mesenchymal stem cells (MSCs) to a certain cell lineage [10,11,12]. In this context, surface microstructuring technologies play a significant role in the manufacturing of 3D scaffolds with modified surfaces, to improve the biocompatibility and performance of these devices to induce tissue regeneration.

Pulsed-laser-based technology is a promising approach for the fabrication of scaffolds to guide the spatial distribution of living cells due to the versatility, precision, and simplicity of this technology for nano-and micro-feature generation on the surface of materials without affecting the properties of the bulk. In 2005 Liu and collaborators applied for the first time an ultra-short pulsed laser for this purpose [13], which reflects the novelty of this strategy. The last years have witnessed the application of the femtosecond-pulsed laser technology to create 3D microstructures on biocompatible materials for cell culture [14,15,16,17,18,19,20,21]. In these studies, channels were created to control distinct aspects of cell behavior on biopolymers. Lee et al. [15] observed that channels fabricated inside electro spun scaffolds made of polycaprolactone favor the formation of vascular walls in that they promote the adhesion of smooth muscle cells on the channel walls. Other authors employed a similar strategy to increase the infiltration of cells and nutrients [16]. Channels carved on a biopolymer film have also been employed to induce cell alignment [17,18] and promote myogenic differentiation of MSCs [18] although the authors did not obtain conclusive results in this respect. Despite the high-quality structures that a femtosecond pulsed laser can generate, it though has technical limitations, such as the low processing speed and the low ablation rate that make it difficult to implement in industry, where both power scaling and versatility are required. In contrast, picosecond pulsed lasers are easier to implement in industrial processes due to their better cost-effectiveness, versatility, and reliability. Consequently, the presence of these lasers has noticeably increased in the industrial market during the last years. Surprisingly, it has been scarcely used for scaffold fabrication [22,23]: previously to our work, only Schlie and collaborators have applied the picosecond laser micromachining technology (PLM) on silicon substrates to test cell compatibility [21], while we employed this technology on biocompatible polymers to create microstructures that modulate the morphology and proliferation of breast cancer cells [23], or enhance the adhesion and promote alignment of human cardiac microvascular endothelial cells [24]. In this work, we examined the effects that carved micropatterns created by PLM on substrates made on Poly-L-Lactide (PLLA) may have on human MSCs and their differentiation into adipocytes and osteoblasts, with the impact that this would have on the fabrication of tailorable and functional scaffolds for cell and tissue engineering, applying a technology that can be used for processing the wide range of biomaterials investigated for tissue engineering. In addition, laser-technologies can be scaled up for the fabrication of 3D components without restrictions in the patterned area or form, contrary to other technologies; such as soft-lithography-based methods [25] with high accuracy in the created feature sizes but which are generally expensive, restricted to a short variety of materials, and difficult to adapt for structuring or patterning non-flat and large 3D components.

## 2. Materials and Methods

### 2.1. Materials

PLLA was supplied by Biomer (Biomer, Krailling, Germany). PLLA sheets of approximately 300 µm in thickness and a degree of crystallinity of 4% (measured by differential scanning calorimetry) were obtained by thermoforming. Under these conditions, the PLLA film is hydrophobic (contact angle = 70° ± 5°) with a surface free energy of 38.18 mJ/m^2^ in air at 23 °C [26].

### 2.2. Surface Microstructuring Technique

Surface microstructuring of the PLLA films was carried out by laser ablation with a picosecond-pulse Nd:YVO4 laser (RAPID, Coherent, Kaiserslautern, Germany) integrated in a micromachining workstation by 3D-Micromac. Laser ablation is the removal of material from a solid surface by direct absorption of laser energy. A detailed description of the experimental set-up and the optimization of the microstructuring process for PLLA can be found in our previous publications [23,27]. 

To analyze the effect of surface roughness on cell activity, flat PLLA substrates (FLAT PLLA) of *Ra* = 240 nm (*Ra*, average surface roughness) were laser-irradiated by UV wavelength (*λ* = 355 nm) applying an energy of 0.9 μJ at a frequency (*f*) of 100 kHz, and 5 µm of pulse distance (*dp*). After this treatment, *Ra* increased to 700 nm (ROUGH PLLA, Figure 1a). To evaluate the effect of surface patterning on the cellular behavior, parallel grooves (width, *w* = 10 μm, and depth, *d* = 4 μm) (Figure 1b) were obtained by applying an energy of 6 µJ at a frequency of 250 kHz, and a pulse distance of 2.4 µm. These groove dimensions matched to human mesenchymal stem cell size (10–12 µm in diameter). The inter-groove spacing was set to 15 µm (GROOVES 1) and 25 µm (GROOVES 2). To analyze the effect of surface microstructure geometry on stem cell growth and differentiation, 3D microcavities were fabricated in different geometrical shapes, such as circles and rectangles (Figure 1c,d), with same parameters (*λ*, *f*, and *dp*) and an energy of 1 µJ. The diameter or side of these geometries is about 200 µm. To ensure cell confinement, the depth of these microcavities was set to 40 µm, approximately 8 times as high as a cell height. Width and depth of the microcavities and grooves were measured by a mechanical stylus profilometer (Dektak 8, Veeco, Plainview, NY, USA). This equipment was also used to measure the Ra on 4 mm long surface profiles, according to DIN EN ISO 4288:1998. The material response on laser—machined regions was assessed using scanning electron microscopy SEM (Karl Zeiss XB1540, Jena, Germany), Focused Ion Beam processing (FIB, FEI Helios 600 Nanolab dual beam focused ion beam system, FEI, Hillsboro, OR, USA), X-ray photoelectron spectroscopy (XPS, hemispherical analyzer PHOIBOS 150 SPECS with 2D-DLD detector, Berlin, Germany), and Fourier transform infrared (FTIR) spectroscopy (Vertex 70 with Hyperion IR microscope, Bruker, Madrid, Spain). The information depths of these techniques are 10 nanometers (XPS) and 1 micrometer (FTIR), respectively.

### 2.3. Cell Culture

Human MSCs from bone marrow were purchased from Promocell (Heidelberg, Germany). Prior to cell culture, all PLLA surfaces were gently cleaned with 70% ethanol and UV-sterilized for 30 min. Cells were cultured on the surfaces in growth medium (Promocell) and maintained at 37 °C and 5% CO_2_. MTT [3-(4,5-dimethylthiazol-2-yl)-2,5-diphenyltetrazolium bromide] assay (Sigma-Aldrich, Munich, Germany) was used to determine the rate of cell viability on all PLLA surfaces (FLAT PLLA, ROUGH PLLA, GROOVES 1, GROOVES 2). The colorimetric assays analyze the number of viable cells since the amount of formazan dye formed directly correlates to the number of metabolically active cells in the culture. MSCs cells were seeded in 24-well plates at a density of 20,000 cells/cm^2^. Medium was changed after 3 days and absorbance was measured after 1, 4, 7, and 14 days. After addition of 50 μL MTT (5 mg/mL) to each well, the mixture was incubated for 4 h, the liquid was removed, 200 μL of dimethyl sulfoxide were then added to each well and the absorbance was read with a UV SpectramaxM2 reader (Molecular Devices) at 550 nm. The relative cell viability was expressed as fold change with respect to cells that were cultured on FLAT PLLA surfaces for one day. These experiments were done in triplicate on three surfaces of each type. Glass coverslips were used as control. Cell morphology was examined on all PLLA surfaces by immunofluorescence microscopy applying two different staining methods: cells were stained with NeuroDio, a green-fluorescent cytoplasmic membrane stain (Promokine, Promocell, Heidelberg, Germany) before seeding and, on the other hand, cells were fixed and stained for DAPI (Vector Biolabs, H-1200, Malvern, PA, USA) (cell nuclei, blue), phalloidin (Thermo Fisher Scientific, A12381, Darmstadt, Germany) (cell cytoplasm, red), and monoclonal anti-vinculin antibody (Sigma, V9131, Sigma-Aldrich, Munich, Germany) (focal adhesions, green). Cell density was 1000 cells/cm^2^ for both staining methods. These experiments were performed in triplicate for every type of surface. Cell morphology was observed at three different time points. 

To study cell differentiation, cells were cultured in adipogenic and osteogenic media provided by Promocell (Heidelberg, Germany), following the instructions of the manufacturer, without addition of any other reagent to the media. MSCs were seeded on PLLA dishes at high density (30,000 cells/cm^2^) to attain 100% confluence after 24 h in culture. The growth medium was then replaced with a differentiation-inducting medium, which contained a 1:1 mix of adipogenic and osteogenic induction media. Cells were cultured in this medium for two weeks. The induction medium was changed every three days. After two weeks, cells were fixed in 4% formaldehyde for 5 min at room temperature and stained immediately with Fast Blue RR Salt/Napthol solution (Sigma-Aldrich, Munich, Germany) and Oil Red O solution (Sigma-Aldrich, Munich, Germany), according to the manufacturer’s instructions. The first staining agent tags alkaline phosphatase activity (AP staining), an early indicator of cells that undergo osteogenesis, while the second tags the deposits of fat or lipid vacuoles characteristic of adipogenesis. Cells were visualized with an inverted microscope in bright field and fluorescence modes (Nikon Eclipse TE-2000-S, 6V30W halogen lamp, bright field (BF), and GFP filter, Tokyo, Japan). 

### 2.4. Quantitative RT-PCR (qPCR)

Real time quantitative PCR (qPCR) was used to analyze expression levels of tissue specific genes: Adiponectin (ADIPOQ) and CCAAT/enhancer binding protein alpha (CEBPα) genes, exclusively expressed in adipose tissue, and runt related transcription factor 2 (RUNX2) essential for osteoblastic differentiation and bone morphogenetic protein 2 (BMP2) genes. 36B4 gene was used as a reference transcript for normalization. Total RNA was isolated using RNA Micro Kit PureLinkTM (Invitrogen), following the instructions of the manufacturer and then analyzed as previously described [28]. The results are presented as fold change calculated with the 2-ΔΔct method. The specific qPCR primers for each marker are as follows: ADIPOQ forward: 5′-AACATGCCCATTCGCTTTACC-3′ and ADIPOQ reverse: 5′-TAGGCAAAGTAGTACAGCCCA-3′; CEBPα forward: 5′-TGGACAAGAACAGCAACGAG-3′ and CEBPα reverse: 5′-TCACTGGTCAACTCCAGCAC-3′; RUNX2 forward: 5′-TCACTACCAGCCACCGAGAC-3′ and RUNX2 reverse: 5′-ACGCCATAGTCCCTCCTTTT-3′; BMP2 forward: 5′-TTTCAATGGACGTGTCCCCG-3′ and BMP2 reverse: 5′-GCAGCAACGCTAGAAGACAG-3′; 36B4 forward: 5′-GTGTTCGACAATGGCAGCAT-3′ and 36B4 reverse: 5′-GACACCCTCCAGGAAGCGA-3′. These experiments were done in triplicate (*n* = 8).

### 2.5. Data Analysis

XPS measurements were performed and analyzed by software “Avantage” from Thermo Fisher Scientific (Darmstadt, Germany). Optical micrographs were analyzed with the image analysis freeware Image J (http://imagej.nih.gov/ij/). Image brightness and contrast were adjusted to optimize the visualization of single cells from a strong light-scattering background. The location and number of clusters of lipid vacuoles inside and outside of grooves were obtained from bright field images of stained MSCs differentiated into adipocytes and osteoblasts, taken after two weeks in culture and at 10 different sample locations. All data were expressed as mean ± standard deviation. To detect whether a significant difference existed among samples, statistical analysis was carried out using Student’s *t*-test and the values were considered significantly different when *p* < 0.05.

## 3. Results

### 3.1. Effect of Laser Irradiation on Material Surface Properties and Microstructure

Grooves like those shown in Figure 1b were machined on amorphous PLLA with a period of 25 micrometers, filling an area of 40 × 3 mm. In these conditions the inter-groove spacing (s) was 8 micrometers. To analyze the effect of laser irradiation on material surface, XPS was applied to obtain the carbon (C1s) and oxygen (O1s) spectra on the grooved area (Figure 2), considering a field of view of 1 × 20 mm. Therefore, the XPS signal has the contributions from two different areas: grooves and not machined inter-groove spacing. Although this 8-µm spacing was not under direct laser irradiation, it was otherwise modified because of the formation of recast material at grooves sides (Figure 1b). In these conditions, the C:O ratio in the pristine (not machined) area (1.9) was higher than the stoichiometric ratio for PLLA (1.5), while the obtained C:O ratio on the grooved area was very similar to the expected value (1.56) (Table 1). The relative intensity of the C1s and O1s peaks—which correspond to the bonding energies of the C–O, C=O, and O–C–=O functional groups—underwent a minor but reproducible change on the grooved area in comparison with the pristine area (Figure 2). Atomic concentrations (at %) of carbon in the C–O and O–C=O functional groups were lower in the grooves, in comparison with the pristine area, while the atomic concentration of carbon in the C=O functional group was increased. These observed differences between pristine and grooved areas were not detected by FTIR measurements (Figure 3). FTIR spectra of both grooved and pristine areas showed a sharp peak arising at 1748 cm^−1^ in the regime of carbonyl stretching. The appearance of this peak at a lower wavenumber region than that of the crystalline structure, and the weak shoulders arising next to the intense signals at 1177 cm^−1^ and 1085 cm^−1^, assigned as asymmetric vibrations of C–CO–O and O–C–CO, respectively, [29] indicate that PLLA exists mainly as amorphous.

In addition to surface chemistry, microstructural properties could also be affected by the energy absorbed by the material under laser ablation, which, if occurring via thermal effects (photothermal ablation) [27], could produce crystallization, not only in the ablated area, but also in the surrounding material. To evaluate whether these eventual structural changes occurred on the laser ablated regions, single pulse craters produced in amorphous PLLA were considered. Cross-sections of non-ablated surface (pristine area) and craters were obtained by FIB and analyzed by SEM (Figure 4). The material response to FIB was similar when considering pristine and crater areas, and no signs of crystallization (for instance, spherulite apparition) were observed on the material close to the ablated zone (Figure 4b,c).

### 3.2. Effect of Surface Topography on Undifferentiated MSCs

To determine the effect of surface topography, cell proliferation was examined on four distinct types of surfaces: FLAT PLLA, ROUGH PLLA, and two different groove-configurations (GROOVES 1 and GROOVES 2) based on grooves with the same dimensions but different inter-groove spacing. Figure 5 shows the relative cell viability on these surfaces as a function of the culture time, measured as the number of cells per area (cells/cm^2^) normalized to cell growth on FLAT PLLA at one day of incubation. Importantly, although cell viability was slightly lower on FLAT PLLA in comparison with glass coverslips (see Appendix A), no significant differences among the different PLLA surfaces were detected.

The PLM fabrication of grooves on PLLA produced two major transformations on the material surface: depressions caused by material removal by the laser pulses and protrusions of recast material at the groove ends and edges (Figure 6). The depth of the depression and the pile of recast material were particularly prominent at the end of the trench (Figure 6a,b). Here, protrusions could be 226% higher than those at any other location along the trench. This phenomenon is ever present in the laser manufacturing technology and it is a consequence of the first-pulse effect [30]. The highly intense first laser pulse that initiates the micromachining process produces a particularly strong effect on the material. 

Figure 7 shows representative images of cell morphology obtained by immunofluorescence confocal microscopy on four different PLLA surfaces (FLAT PLLA, ROUGH PLLA, GROOVES 1, and GROOVES 2) using cover slips as control. Immunofluorescent staining was used to examine cell cytoskeleton (phalloidin) and nuclei (DAPI) and focal adhesion points (vinculin) on these surfaces. Cell proliferation and adhesion were lower on the PLLA surfaces than on standard glass cover slips. Analysis of cell morphology showed cell alignment, both in terms of cell cytoplasm and nuclei, along the groove-patterned surfaces. In addition, substrate topography also affected cell nuclei orientation (Appendix A).

Contact guidance of MSCs on grooves occurred within the patterned areas (Figure 7d,e). However, in the regions between grooves, and within the first 24 h in culture, cells spread across the gap, extending between groove endpoints of neighboring patterned areas (Figure 8a), or between endpoints and groove edges (Figure 8b). These findings suggest that single cells develop filopodia-like extensions between topological protrusions, which in this case are at least 200 µm apart.

### 3.3. Effect of Surface Topography on Differentiated Human MSCs

In vitro differentiation of MSCs is most efficient when it is induced at high cell confluency (i.e., 90%). Under these conditions, it has been suggested that cell-cell interactions may be more relevant than cell-substrate interactions to determine shape and orientation of MSCs [31]. Nevertheless, we wished to explore whether the presence of topological barriers and cavities may alter the behavior of differentiated MSCs (i.e., adipocytes and osteoblasts). Adipocytes and osteoblasts descend from a MSC precursor. Adipocytes are round to allow maximal lipid storage in the adipose tissue, while osteoblasts tend to spread to facilitate matrix deposition activity [31].

After 14 days of adipogenic differentiation, the distribution of lipid vacuoles cultured on FLAT and ROUGH PLLA was examined. In both cases, the clusters of lipid vacuoles were quasi globular, and their distribution was not affected by substrate roughness (Figure 9a,b). Similarly, AP staining suggested that osteoblast distribution was unaffected. However, the lipid vacuoles appeared to follow the edge separating regions of different roughness (Figure 9c). A similar behavior was observed on the edges of the microcavities, where cells confronted a topological barrier of approximately 40 µm in height (Figure 9d,e). Lipid vacuoles close to the edge of these cavities followed the borderline regardless of the shape and size of the microcavity, which in this case was much larger than a single cell. On grooved-patterned PLLA, the distribution of the lipid vacuoles on the surface differed (Figure 10a–c): Lipid vacuoles arranged in strings both inside and in between the grooves, however, they were more likely to be found inside the grooves (67 ± 11%) than outside (29 ± 13%) (Figure 10d). In contrast, osteoblasts did not show any noticeable predefined orientation on the patterned surface. 

To determine MSC differentiation on different surfaces, the expression levels of two adipogenic (ADIPO Q and CEBPα) and two osteogenic (RUNX2 and BMP2) specific genes on rough and grooved surfaces compared to flat PLLA were analyzed. Gene expression levels were measured on these surfaces considering four distinct culture conditions: growth medium (normal MSC culture medium), 100% adipogenic differentiation induction medium (Adipo), 100% osteogenic differentiation induction medium (Osteo) and 1:1 adipogenesis:osteogenesis induction medium (Mix). Analysis of adipogenic markers indicated that MSCs differentiated into adipocytes only when cultured in adipogenic or mixed media, however, no significant effect of surface patterns on cell differentiation into adipocytes was observed. A modest, although not statistically significant, increase of expression levels of these markers was detected in cells differentiating on grooves under adipogenic medium (Figure 11a,b). In contrast, a significant increase in osteogenic marker expression was observed when cells were cultured in normal medium on patterns in comparison with FLAT PLLA. In addition, differentiation media did not further increase osteogenic marker expression on the patterned surfaces (Figure 11c,d). These findings suggest that alterations on PLLA surface topography are enough to induce osteogenesis, independently of culture medium conditions. Furthermore, osteogenesis was favored over adipogenesis for all four surfaces (including FLAT PLLA) when cultured in mixed induction medium (Figure 11e). 

## 4. Discussion

We have examined the effect of picosecond pulsed laser ablation on surface properties of PLLA films. The C1s and O1s spectra obtained by XPS at grooves showed minor changes on the C:O ratio and the relative intensity of the C–O, O–C=O, and C=O components with respect to the not machined surface (Figure 3). The high C:O ratio measured on the not machined PLLA surface in comparison with the theoretical value has already been observed by other authors, connecting this finding to the sensitivity of obtained ratio to the take-off angle at which photoelectrons are recorded as well as the segregation of methyl groups at the surface [32,33]. The C:O ratio in the grooved area is very similar to the stochiometric ratio, indicating that no photo-oxidation occurred, as has been observed when femtosecond laser ablation is applied on PLA [34]. The minor changes found on the relative intensity of the above-mentioned functional groups on the grooved area in comparison with the non-modified area indicate a re-organization of these functional groups on the polymer surfaces. This could be a sign of photodecomposition and, therefore, of the photochemical nature of the ablation mechanism, at least in part. In addition, these surface changes were not observed by FTIR (Figure 3), indicating that only the topmost surface layer of the material (less than 10 nm) is affected by the laser and no bulk properties are modified. It is worth to note that no crystallization was detected neither by FTIR nor SEM analysis of FIB-obtained cross sections, and no structural changes were found on the material around the ablated area. The same has been observed by Shibata et al. [35] on PLGA films irradiated by femtosecond laser pulses and characterized by XRD (X-Ray diffraction).

We have also analyzed the effects that PLLA substrates modified by picosecond pulsed laser ablation have on MSC shape, and differentiation into adipocytes and osteoblasts. MSCs cultured on patterned PLLA showed the characteristic contact guidance effect [36], both in terms of cell cytoplasm and cell nucleus, already after 24 h in culture and throughout a period of 14 days. These results agree with previous reports on stem cell alignment along micro- and nano-grooved-patterned substrates [19,37,38,39]. Cell nuclei alignment is promoted on grooves with minor inter-groove spacing in comparison with the further apart grooves, with an increment of approximately 25% (see Appendix A). A minor inter-groove spacing also increases the percentage of cell nuclei confined inside the grooves: 85% of cell nuclei were found attached to the inner surface of the grooves for *s* = 15 µm, while 54% of cell nuclei were attached in the spacings between grooves for *s* = 25 µm. One possible explanation for this behavior could be the slight change undergone by the topographical profile of the inter-groove spacing when this increases (Figure 12). For an inter-groove spacing of 15 µm, the surface between grooves is formed by the merger of the recast material deposited at both sides of each groove, leading to a spacing surface that is approximately flat. However, for an inter-groove spacing of 25 µm, groove ridges formed by the recast material at both sides of each groove separate from each other, leading to the formation of a shallow channel between the grooves, which is wide enough to have the same effect as a ‘proper’ groove and promote cell nuclei confinement. In addition, the protrusions generated by the laser ablation technique at grooves’ ends and grooves’ edges provide convenient anchorage points for MSC and can potentially control their adhesion and shape. Hamilton and collaborators observed a similar phenomenon in osteoblasts proliferating on boxes and pillars [40], which they called gap guidance, a type of contact guidance for cell alignment that is associated to discontinuous topographical edges. Our results on microgrooves show that these microstructures influence MSCs in two ways: on the one hand, grooves influence cell orientation, since cells adapted their shape to groove width (in terms of cell cytoplasm) and orientation (both cell cytoplasm and nucleus), and the maximal effect was observed at the first stages of MSCs proliferation; on the other hand, grooves’ edges promote cell adherence and provide guidance. These effects appear to resemble the influence of stem cell niches in vivo [5], which makes PLM a convenient technique to create in vivo-like cellular environments.

In terms of MSCs differentiation, the data show that adipocytes, in contrast to osteoblasts, are highly sensitive to topographical features. Lipid vacuoles in adipocytes cultured on patterned PLLA align in strings along the grooves. Similarly, the distribution of lipid vacuoles was affected by the presence of topological edges, such as the borderline between flat and rough PLLA, as well as the walls of microcavities. The images of adipocytes on non-modified substrates suggest that the adipocyte shape is defined by the distribution of the lipid vacuoles. According to this, it can be expected that when lipid vacuoles arrange in lines of beads alongside the grooves, the adipocyte bodies may as well be aligned with these grooves (Figure 13). To the best of our knowledge, this is the first observation of adipocyte compliance to exclusively topological cues in the micrometer scale. Kim et al. [41] reported on preadipocyte alignment, in terms of cell cytoplasm, on nanometric grooves fabricated on polyurethane acrylate surfaces treated by oxygen plasma (to make the surface hydrophilic) and coated by fibronectin to increase cell adhesions. However, they did not observe the confinement and alignment of the lipid vacuoles inside the grooves, and the shape of the adipocytes looked very similar on the nanogrooves in respect to the control (non-patterned) surface (adipocytes showed circular distribution of lipid vacuoles around the cell nucleus). Our finding reveals that surface topography alone can control adipocyte morphology. Mature adipocytes release a wide variety of factors that play a fundamental role in the regulation of many essential functions of the body [42]. The expression of these factors is, in many cases, controlled by adipocyte size and location [43,44]. In this context, adipocytes should be able to change their morphology to store the optimum amount of fat and to perform properly their physiological functions [45]. Therefore, it is likely that, according to the results presented in our study, surface topography could also affect the expression of these factors as it influences adipocyte morphology. Hence, PLM of surfaces may greatly contribute to nano- and regenerative medicine, since surface topography could be tailored to treat diseases related to the dysfunctional expression of those factors. In addition, specially-designed scaffolds can be employed to promote adipocyte adhesion and direct the formation of adipose tissue for repairing soft tissue defects [46], or for implantation of artificial organs that require fat tissue regeneration into a predefined geometry, such as an ear or larynx [47,48].

Several authors have reported on the effect of the pattern geometry on cell differentiation into a certain lineage based on biochemical cues, as it is the case for surfaces patterned with agents that induce or hinder cell adhesion [7,49,50,51,52]. In these studies, chemical restrictions were imposed on stem cells, which induce mechanical force gradients at the pattern edges that lead to differentiation into a particular cell type. In our research, where cells were confined by purely physical means with no changes on chemical or mechanical surface properties, we found no major differences in the spatial distribution of osteoblasts and adipocytes inside the microcavities in comparison with the non-patterned material (see Figure 9 and Figure 10). However, osteogenesis was predominant on groove and rough patterns when cells were cultured in mixed osteogenesis/adipogenesis induction medium (see Figure 11e). In addition, a small, but significant, increase of osteogenic expression levels was observed on the patterns, particularly on grooves, with respect to the non-patterned surface when cells were cultured in growth (normal) medium (Figure 11c,d). The grooved patterns alone were able to promote osteogenesis at the same or even higher level than the induction agents of the osteogenic differentiation induction medium. The induction of osteogenic differentiation by surface patterns has been observed by other authors when applying specific features both in the nanoscale and the microscale (pits, pillars, or grooves) for MSCs cultured in vitro under both inductive and non-inductive medium [53,54,55,56,57]. Those studies state that the osteogenesis induction promoted by topological cues may be related to large focal adhesions, enhanced cell areas, and well-organized cytoskeleton. However, there is still controversy as to whether flattened [53] or elongated [56] morphologies are preferred for osteogenesis and whether a biochemical stimulus is needed to induce cellular differentiation. Regarding the influence of cell morphology on cellular differentiation, our study supports the observations found by Kim et al. [56] related to a maximized Ad-MSCs differentiation to osteoblasts for cells aligned along the grooves and which show an increase in the cell length but none or little in width in comparison with the flat surface. In relation to the need of induction agents for cell differentiation, our findings prove that osteogenesis can be significantly induced by surface topography alone, in accordance with the observations of Watari et al. [53] about promotion of osteogenic differentiation of hMSCs cultured on nanogrooves without inductions agents. Contrary to the results found in the above-mentioned studies, our results show that the considered patterns do not produce a significant increase on cell differentiation compared to FLAT PLLA under differentiation induction media. However, it is important to point out that the differentiation pathway of MSCs will be highly dependent on the culture medium, the biochemical properties of the material, and the initial cell seeding density [53,56]. In addition, it is worth noting that the mix differentiation medium seems to offer enough induction agents for the cells to undergo osteogenesis (both on flat and patterns) at the same extent as the pure osteogenesis induction medium did. Regarding differentiation, the topography of the PLLA surface seems to have a bigger influence on osteogenic differentiation than the type of culture medium (mix or pure osteogenesis induction medium).

It is worth highlighting that all the cellular behaviors reported here were controlled only by physical/topological mechanisms at the micro-scale, without the interplay of chemical factors, and therefore, without modification of the chemical surface properties. This would enable to control independently both physical and chemical surface properties of scaffolds to get a versatile surface properties “palette” which could be customized to each biomaterial and biomedical application. In addition, picosecond lasers can reach high mechanizing speeds, leading to faster processing than other microfabrication technologies, which makes it highly convenient for processing and manufacturing of large and complex 3D plastic components, like those required for scaffold fabrication. Laser micro-processing offers many advantages compared to the other existing surface modification technologies, such as versatility in terms of materials to be processed and geometries to be generated, the fact of presenting a single-step and contactless method, and a simple adaptation of the process for micropatterning of more complex shapes.

## 5. Conclusions

The present study shows that substrate topography at the micrometer scale affects the morphology, orientation, and differentiation of human mesenchymal stem cells (MSCs) without the interplay of biochemical factors. Laser-generated microstructures induce contact guidance of MSCs, favoring cell organization and directing cell anchorage. The cellular distribution of lipid vacuoles is aligned along topological edges and confining barriers, indicating that adipocytes are influenced by substrate topography. Furthermore, surface topography alone is enough to promote MSC differentiation into osteoblasts, independently of medium conditions. These findings suggest that surface topography impacts on the biocompatibility and functionality of biomaterials that can be used as scaffolds in tissue engineering and/or in vitro studies of MSC behavior under specific physical constraints. In addition, these topographical cues are precisely and simply generated by a high-efficiency technology with low processing times compared to conventional micropatterning techniques such as lithography-based techniques. In conclusion, PLM represents a potential tool for surface modification of scaffolds which may mimic the stem cell niche to control or guide cell organization and differentiation.

## Figures and Tables

**Figure 1 polymers-10-01337-f001:**
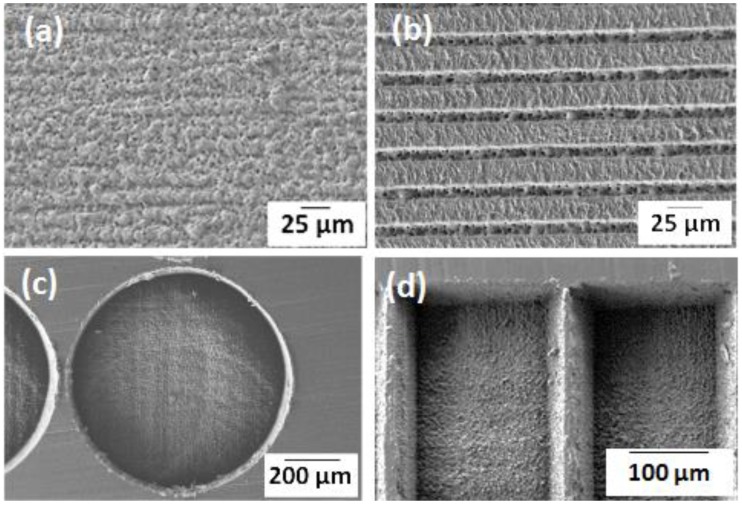
SEM images of laser-treated PLLA: laser-irradiated (“rough”) PLLA; (**a**) grooves with 10 μm width, 4 μm depth, and 15 μm spacing; (**b**)circular; (**c**) and rectangular; (**d**) microcavities.

**Figure 2 polymers-10-01337-f002:**
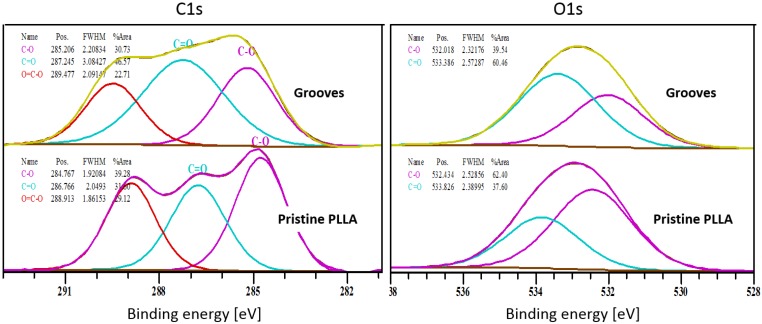
C1s and O1s spectra obtained by XPS on grooved and pristine areas of amorphous PLLA.

**Figure 3 polymers-10-01337-f003:**
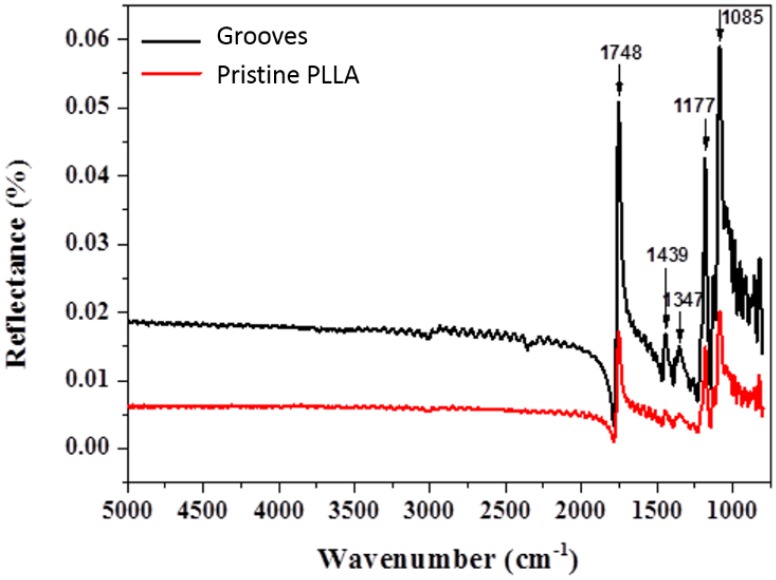
Infrared spectra of grooved and pristine areas of an amorphous PLLA film. Black and red lines represent spectra of grooved and pristine PLLA areas respectively.

**Figure 4 polymers-10-01337-f004:**
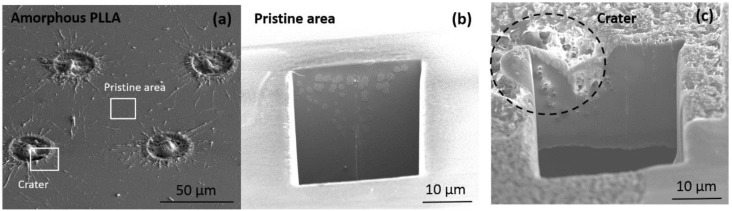
SEM images of craters produced by UV wavelength on amorphous PLLA; (**a**) and cross sections obtained by FIB on the non-ablated area (pristine, (**b**)) and craters (**c**).

**Figure 5 polymers-10-01337-f005:**
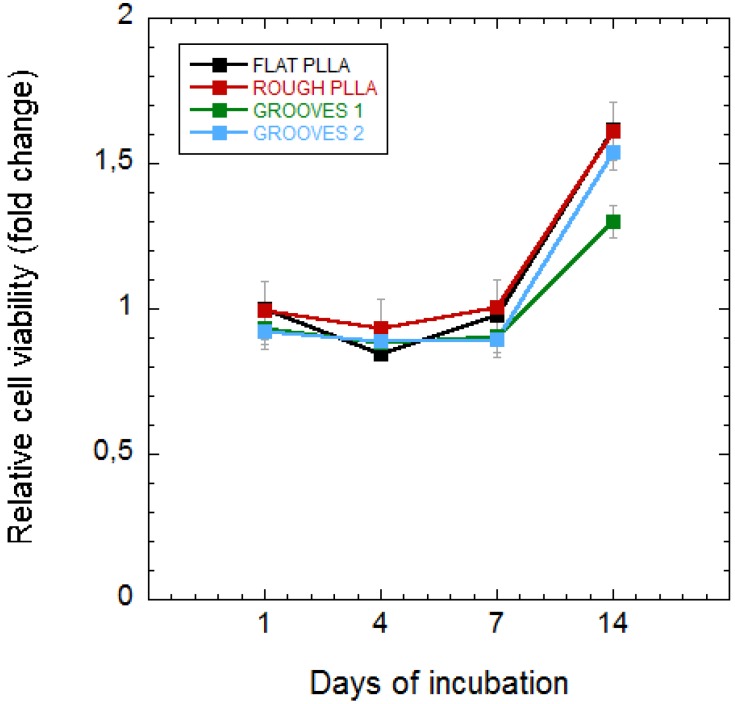
MSC proliferation (cells/cm^2^), as quantified by MTT assays, on four types of surfaces and at different incubation times. Cell growth was normalized to the absorbance obtained with cells cultured on FLAT PLLA after one day of incubation, which was set as 1. The types of surfaces included: PLLA treated by laser to increase the surface roughness (ROUGH PLLA), and PLLA patterned by grooves of width = 10 µm, depth = 4 µm, and spacing = 15 µm (GROOVES 1) and 25 µm (GROOVES 2) in comparison with FLAT PLLA (b).

**Figure 6 polymers-10-01337-f006:**
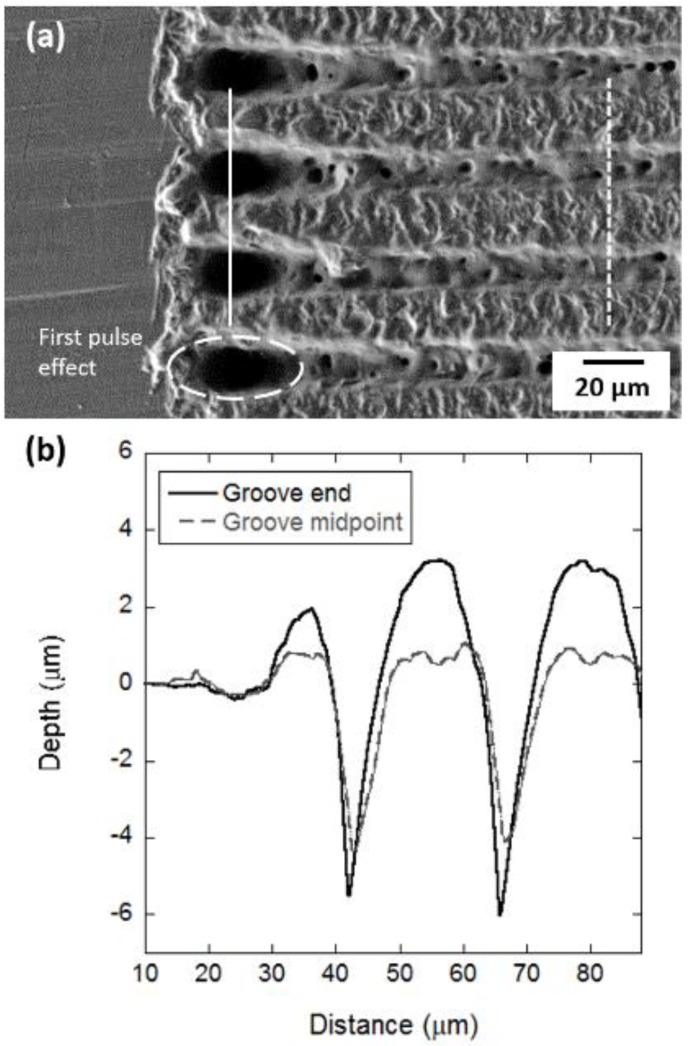
SEM image of grooves created by picosecond pulsed laser ablation; (**a**) line profiles across the grooves: at the end of the trenches (continuous line) and at the midpoint of the trenches (dashed line) (**b**).

**Figure 7 polymers-10-01337-f007:**
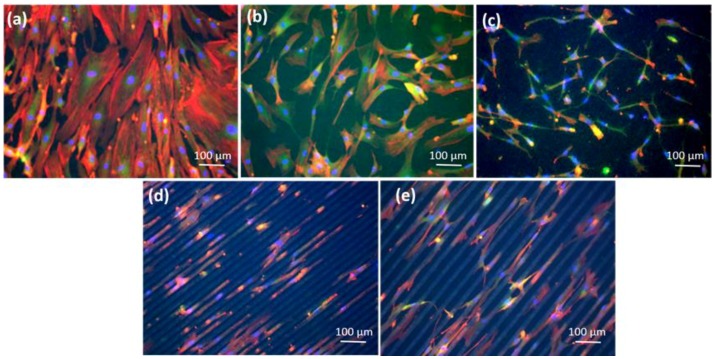
Representative immunofluorescence images of MSCs on glass cover slips or on four different PLLA surfaces after 14 days in culture: (**a**) MSCs on cover slips; (**b**) MSCs on FLAT PLLA; (**c**) MSCs on ROUGH PLLA; (**d**) MSCs on GROOVES 1; (**e**) MSCs on GROOVES 2. Focal adhesions marked with Vinculin are shown in green, cell cytoplasm marked with Phalloidin in red and cell nuclei marked with DAPI in blue.

**Figure 8 polymers-10-01337-f008:**
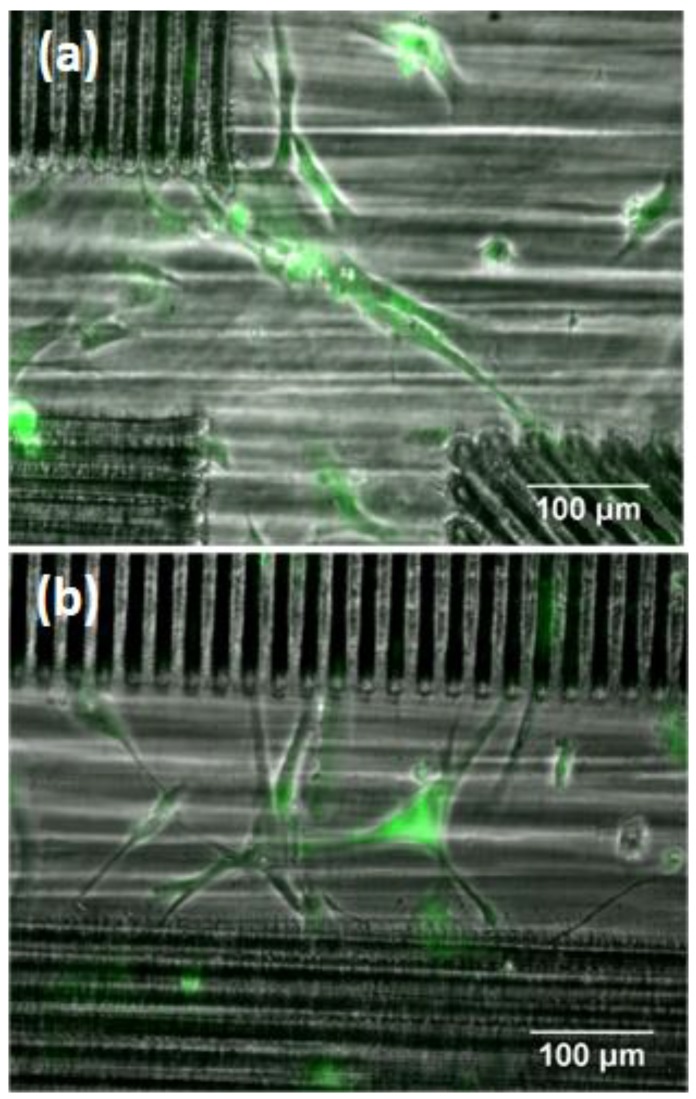
Human MSCs adhered on flat PLLA in between laser-patterned squares after 24 h in culture. Cell spreading from one groove endpoint to another of the second-nearest neighbor square (**a**), and from groove endpoints to groove edges of a first-nearest neighbor square (**b**). Images were treated by ImageJ to highlight cell morphology (green color).

**Figure 9 polymers-10-01337-f009:**
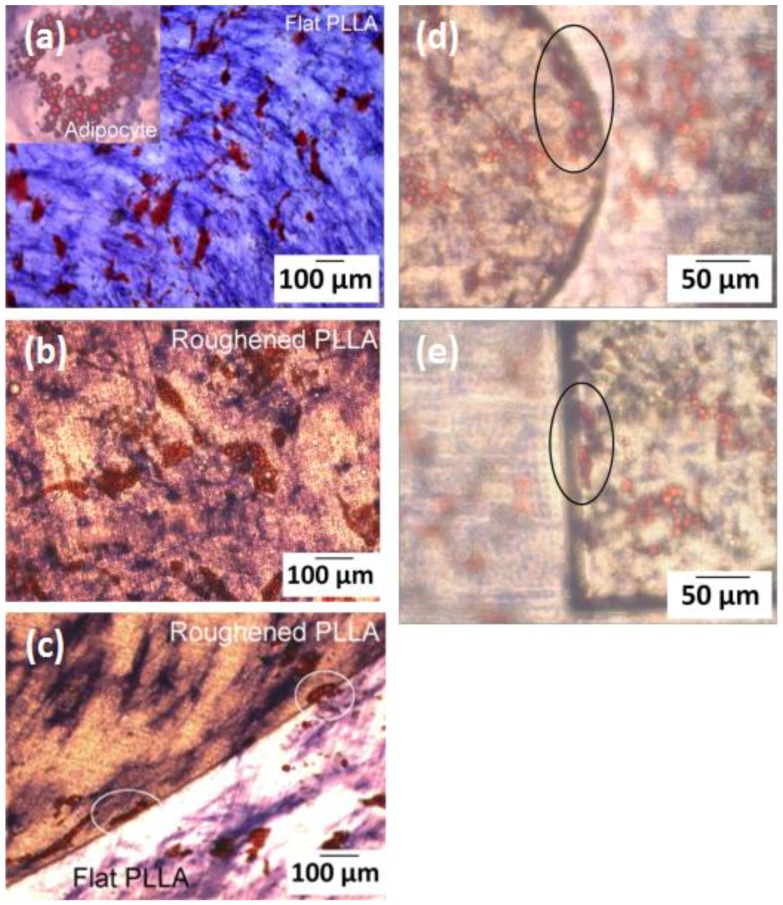
Bright field (BF) images of MSCs differentiated into adipocytes (presence of lipid vacuoles in red) and osteoblasts (AP staining in blue) on FLAT PLLA; (**a**), ROUGH PLLA; (**b**,**c**), and in microcavities (**d**,**e**). In (**c**)–(**e**), strings of lipid vacuoles following the walls of the microcavities are outlined by white and black ovals, respectively.

**Figure 10 polymers-10-01337-f010:**
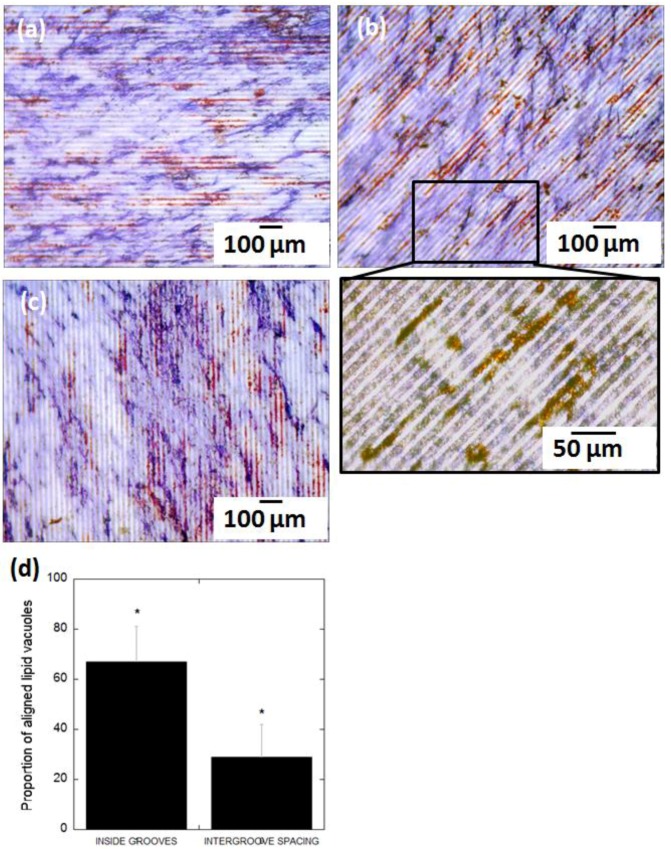
BF images of MSCs differentiated into adipocytes (presence of lipid vacuoles in red) and osteoblasts (AP staining in blue) on GROOVES after 14 days in culture (**a**–**c**); (**d**) Ratio of clusters of lipid vacuoles confined in the grooves and on the inter-groove spacing (* significance level according to the Student *t*-test: *p* < 0.005).

**Figure 11 polymers-10-01337-f011:**
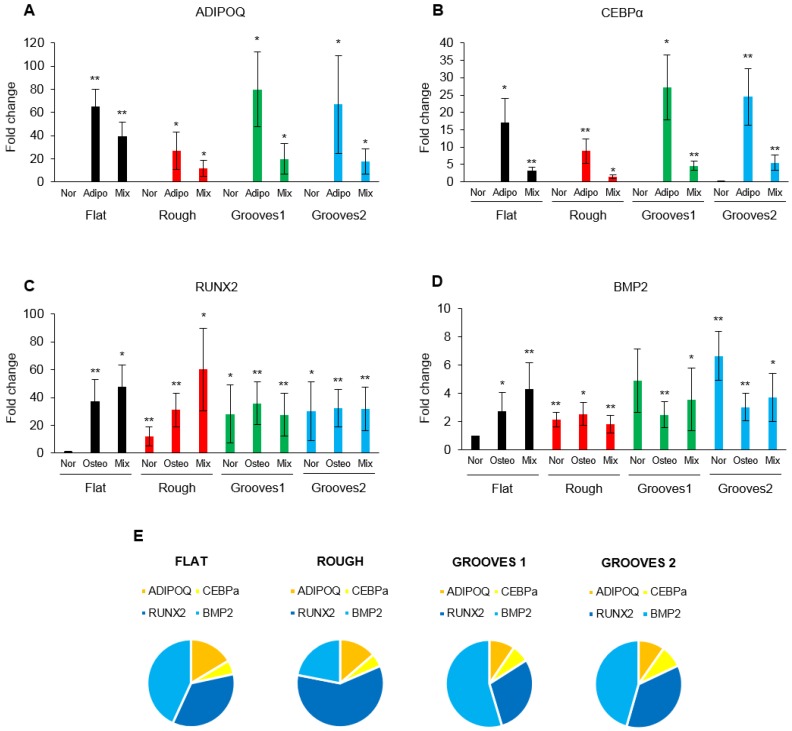
mRNA expression levels of tissue specific genes normalized to an endogenous control and the FLAT PLLA control: Adiponectin (ADIPOQ); (**a**) exclusively expressed in adipose tissue, and CCAAT/enhancer binding protein alpha (CEBPα) genes; (**b**), and runt related transcription factor 2 (RUNX2); (**c**) essential for osteoblastic differentiation and bone morphogenetic protein 2 (BMP2) genes; (**d**) which play a role in bone and cartilage development. Mesenchymal stem cells were cultured on different PLLA surfaces (FLAT, ROUGH, GROOVES 1 and 2) using different media (normal, to maintain undifferentiated cells, adipo, to induce adipogenic differentiation, osteo, to induce osteoblast differentiation and a mix of both induction media 1:1), as indicated (*n* = 5) ** *p* < 0.01 and **p* < 0.05 by *t*-test. Results are representative data from three independent experiments. Pie-graphs of differentiation markers expressed by cells when cultured on the 4 different PLLA surfaces under mix medium; (**e**) normalized to expression levels of each marker under the corresponding pure induction medium (osteogenesis or adipogenesis).

**Figure 12 polymers-10-01337-f012:**
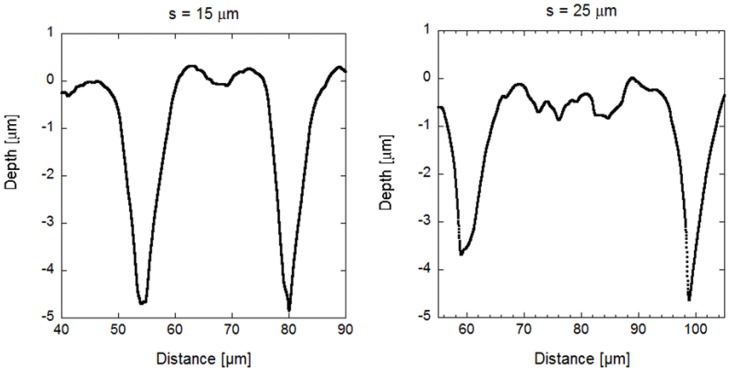
Topographical profile measured by profilometry of the laser-created grooves for an inter-groove spacing of 15 and 25 µm.

**Figure 13 polymers-10-01337-f013:**
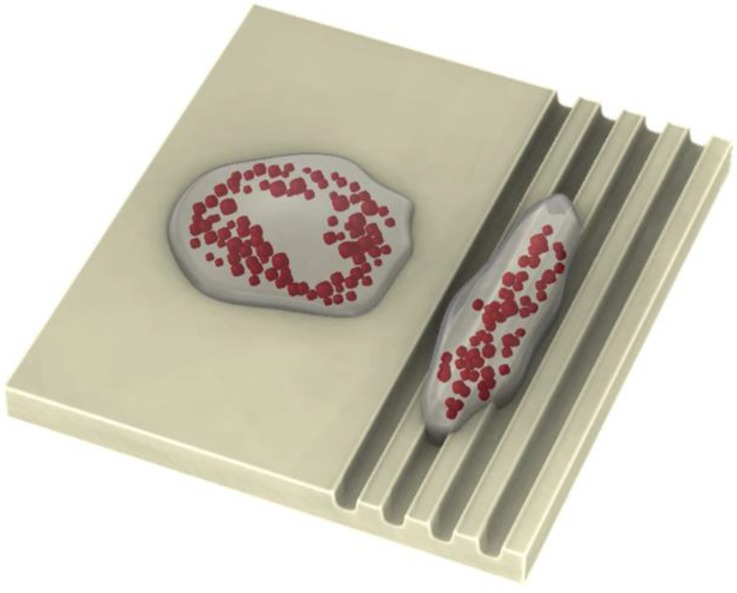
Adipocytes on a flat substrate and on grooves according to the experimental observations.

**Table 1 polymers-10-01337-t001:** C.O ratios (at %) in the pristine and grooved PLLA surfaces measured by XPS.

	Pristine Area		Grooved Area	
	C 1s	O 1s	C 1s	O 1s
Position	285.50	532.50	286.50	533.00
at %	65.61	34.39	60.94	39.06
C:O	1.9		1.56	

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
