# Peer review of "Laser Surface Microstructuring of a Bio-Resorbable Polymer to Anchor Stem Cells, Control Adipocyte Morphology, and Promote Osteogenesis"

_polymers, 2018, doi:10.3390/polym10121337_

Reviewer 1 Report

The authors demonstrated the laser surface microstructuring PLLA film for cell adhesion and growth in this study, and found that the PLM technique is highly effective to fabricate 3D microstructure precisely for controllable cell adhesion and growth, as well as for controlling adipocye morphology and induce osteogenesis without biochemical functionalizing. It is a very interesting work. The manuscript is good-organized and well-written. All the obtained results are convincible. This manuscript is recommended for publication at Polymers after minor revisions.

1) It is suggested for the authors to add more information on the novelty and significance of this work.

2) In Fig. 5, what is the unit of Y axis? how to measure the relative cell growth?

3) In Fig. 6, the scale bar for image a should be added.

4) Is it possible for replacing the PLLA film with other similar polymers for this study? more discussion is needed.

Author Response

Point 1: It is suggested for the authors to add more information on the novelty and significance of this work.

Response 1: The following information has been added in line 80, in the second paragraph of the introduction section:

“In this work, we examined the effects that carved micropatterns created by PLM on substrates made on Poly-L-Lactide (PLLA) may have on human MSCs and their differentiation into adipocytes and osteoblasts, with the impact that this would have on the fabrication of tailorable and functional scaffolds for cell and tissue engineering, applying a technology that can be used for processing the wide range of biomaterials investigated for tissue engineering. In addition, laser-technologies can be scaled up for the fabrication of 3D components without restrictions in the patterned area or form, contrary to other technologies, such as soft-lithography-based methods [25], with high accuracy in the created feature sizes, but generally expensive, restricted to a short variety of materials and difficult to adapt for structuring or patterning non-flat and large 3D components.”

Point 2: In Fig. 5, what is the unit of Y axis? how to measure the relative cell growth?

Response 2: In order to determine whether surface topography affected cell growth, we performed MTT assays. These colorimetric assays analyze the number of viable cells cultured under different conditions. The measure absorbance directly correlates to the number of viable MSC cells and the values obtained from cells cultured on FLAT PLLA at day 1 were considered as a reference and set at 1. The relative cell growth of each condition was normalized with respect to this reference point and it is displayed as fold change. Sections 2.3 (line 133) and 3.2 (line 251), the Y axis label and the caption of this figure have been rephrased to clarify this point:

“Figure 5 shows the relative cell viability on these surfaces as a function of the culture time, measured as the number of cells per area (cells/cm2) normalized compared to cell growth on FLAT PLLA at 1 day of incubation.”

“Figure 5. MSC proliferation (cells/cm2), as quantified by MTT assays, on 4 types of surfaces and at different incubation times. Cell growth was normalized to the absorbance obtained with cells cultured on FLAT PLLA after 1 day of incubation, which was set as 1. The types of surfaces included: PLLA treated by laser to increase the surface roughness (ROUGH PLLA), and PLLA patterned by grooves of width = 10 µm, depth = 4 µm and spacing = 15 µm (GROOVES 1) and 25 µm (GROOVES 2) compared to FLAT PLLA (b).”

Point 3: In Fig. 6, the scale bar for image a should be added.

Response 3: The scale bar for Figure 6 has been added.

Point 4: Is it possible for replacing the PLLA film with other similar polymers for this study? more discussion is needed.

Response 4: The laser technology considered in this study for ablation of the polymer to create different surface topographies can be applied in any type of material including different types of polymers, glass, ceramics and metals. The laser wavelength and energy can be modified for ablation of transparent and not transparent polymers, so this study could be performed considering other polymers. Since the observed cell behavior in terms of alignment, anchoring and differentiation was caused by the surface topography and no chemical changes were produced by the laser on the surface, we can assume that this behavior could be induced in MSCs cultured on other polymers modified by laser ablation to create the same type of topological cues.

The following paragraph was added at the end of the discussion section to discuss this issue, line 499:

“It is worth highlighting that all the cellular behaviors reported here were controlled only by physical/topological mechanisms at the micro-scale, without the interplay of chemical factors, and therefore, without modification of the chemical surface properties. This would enable to control independently both physical and chemical surface properties of scaffolds to get a versatile surface properties “palette” which could be customized to each biomaterial and biomedical application. In addition, picosecond lasers enable to reach high mechanizing speeds leading to faster processing than other microfabrication technologies, which make it highly convenient for processing and manufacturing of large and complex 3D plastic components, like those required for scaffold fabrication. Laser micro-processing offers many advantages compared to the other existing surface modification technologies, such as versatility in terms of materials to be processed and geometries to be generated, the fact of being a single-step and contactless method, and the easy adaptation of the process for micropatterning of more complex sample shapes.”

Reviewer 2 Report

It is a very interesting paper for the investigation of the effect of different topographies by Laser surface microstructuring on polymeric surfaces on the cellular adhesion and differentiation of hMSCs.

1.Recent works and Review articles regarding ultrashort laser processing and the effect of the fabricated topographies on metallic and polymeric surfaces on cell adhesion and differentiation [1,8]  should be referenced at the Introduction section. Furthermore on soft lithography of different topographies on cell adhesion and differentiation (mentioned on the Conclusions) [9,10]. 

2. How did you measure the Ra on the topographies? Only by the Standard? It was not very clear.

3. More details are required on the methodologies. Please describe more the MTT assay, the osteogenic and adipogenic medium. They are provided by Promocell, but you should add more info regarding the ingredients to induce differentiation eg dexamethasone etc.

Line121: Phrasing should be changed. The MTT assay is used to measure cytotoxicity, not cell proliferation. 

Line 148: Was the 36B4 gene the only one that was tested for the normalisation of the data? Or were there more genes that were tested and 36B4 was the one to be chosen as the most stably expressed gene to use for the normalisation?

4. Figure 5: Please explain where is the control group? Is it normalized to all the groups? What is the relative cell growth? Why from 1 to 7 days there is no relative cell growth and then suddenly at 14 days there is a great increase? Maybe the normalization is not efficient. For MTT 20000cells/cm2 is a high cell starting number, depending on the well size. More details on the experiment are required.

5. Figure 7: Higher magnification images would show a clearer focal adhesion view related to vinculin.

6. Section 3.3: MSCs cultured only with adipogenic induction medium do they follow the same pattern? 

7. Figure 9: Images from control groups would be a useful addition in this figure (MSCs with adipogenic induction medium only and MSCs with osteogenic induction medium only but on different substrates) 

8. Figure 11:   It is seems that in the mix medium and on grooves you have more osteoblasts than adipocytes. In those cases how do you do the normalization with the endogenous control (36B4 gene) since you cannot distinguish between 36B4 mRNA form osteoblasts and 36B4 mRNA from adipocytes?      

9. Minor Corrections

Line 54:  “… The last years have witnessed…”

Line 323: “A modest, although not statistically significant…”

Line 341: Change phrasing to better define the composition of the mixed medium.

Line 436: “…, particularly on grooves, in respect to…”

Line 449: “… but none or little in width compared to the flat surface.”

Line 459: “… at the same extent as the pure osteogenesis….

[1] Acta Biomaterialia 6 (2010) 2711–2720

[2] J Tissue Eng Regen Med (2015);9: 424–434

[3] Biomaterials 67, 115-128

[4] Biofabrication 9, 025024 (2017)

[5] Acta Biomaterialia, 51, 21 (2017)

[6] Interface focus, 4(1), 20130048 (2014)

[7] International journal of molecular sciences 19, 2053 (2018)

[8] ChemPhysChem 19, 1143-1163 (2018)

[9] Biofabrication 3, 045004 (2011)

[10] Journal of nanobiotechnology, 12, 60. (2014)

Author Response

Point 1: Recent works and Review articles regarding ultrashort laser processing and the effect of the fabricated topographies on metallic and polymeric surfaces on cell adhesion and differentiation [1,8] should be referenced at the Introduction section. Furthermore on soft lithography of different topographies on cell adhesion and differentiation (mentioned on the Conclusions) [9,10].

Response 1: In order to avoid a too extensive bibliography and help the reader to focus on the topic addressed in this study, we would like to include only references related to direct ablation of biopolymers by ultrashort lasers for scaffold fabrication. Therefore, we have added two of the articles suggested by the reviewer regarding ultrafast laser fabrication of microstructures on PET for controlling the adhesion, proliferation and orientation of Schwann cells [21], and the replication of laser micro/nano structures from silicon molds to a biodegradable polymer by soft-lithography [25], as an example of this technique.

Point 2: How did you measure the Ra on the topographies? Only by the Standard? It was not very clear.

Response 2: The Ra was measured by a mechanical stylus profilometer (Dektak 8, Veeco, USA) applying the mentioned standard. The sentence referred to this, in line 115, has been modified to clarify this issue:

“Width and depth of the microcavities and grooves were measured by a mechanical stylus profilometer (Dektak 8, Veeco, USA). This equipment was also used to measure the Ra on 4 mm long surface profiles, according to DIN EN ISO 4288:1998.”

Point 3: More details are required on the methodologies. Please describe more the MTT assay, the osteogenic and adipogenic medium. They are provided by Promocell, but you should add more info regarding the ingredients to induce differentiation eg dexamethasone etc.

Line121: Phrasing should be changed. The MTT assay is used to measure cytotoxicity, not cell proliferation.

Line 148: Was the 36B4 gene the only one that was tested for the normalisation of the data? Or were there more genes that were tested and 36B4 was the one to be chosen as the most stably expressed gene to use for the normalisation?

Response 3:

MTT assays (Cell Proliferation kit 1 (MTT), as indicated in the Sigma catalog (https://www.sigmaaldrich.com/catalog/product/roche/11465007001?lang=es&region=ES&gclid=EAIaIQobChMIvt-6hrXy3gIVhYbVCh0AoA_kEAAYAiAAEgKJs_D_BwE) is a calorimetric assay for the nonradioactive quantification of cellular proliferation, viability and cytotoxicity. They analyze the number of viable cells since the enzymatic reaction quantified is part of the respiratory chain of the mitochondria, which is only active in metabolically intact cells.

The section 2.3 has been modified to add the following information:

In line 133:

“MTT [3-(4,5-dimethylthiazol-2-yl)-2,5-diphenyltetrazolium bromide] assay (Sigma-Aldrich, Germany) was used to determine the rate of cell viability on all PLLA surfaces (FLAT PLLA, ROUGH PLLA, GROOVES 1, GROOVES 2). The colorimetric assays analyze the number of viable cells since the amount of formazan dye formed directly correlates to the number of metabolically active cells in the culture. MSCs cells were seeded in 24-well plates at a density of 20000 cells/cm2. Medium was changed after 3 days and absorbance was measured after 1, 4, 7 and 14 days. After addition of 50 μl MTT (5 mg/ml) to each well, the mixture was incubated for 4 h, the liquid was removed, 200 μl of dimethyl sulfoxide were then added to each well and the absorbance was read with a UV SpectramaxM2 reader (Molecular Devices) at 550 nm. The relative cell viability was expressed as fold change with respect to cells that were cultured on FLAT PLLA surfaces for 1 day.”

In line 154

“To study cell differentiation, cells were cultured in adipogenic and osteogenic media provided by Promocell (Heidelberg, Germany), following the instructions of the manufacturer, without addition of any other reagent to the media.”

Unfortunately, Promocell was unable to provide any information regarding the composition of the osteogenic and adipogenic induction media. 

The 36B4 cDNA nucleotide sequence has highly conserved regions in the 5-prime end of its open reading frame across different tissues. When compared to the transcript levels of other common reference genes, such as beta-actin and cyclophilin, 36B4 proved to be a very reliable and consistent standard for use in gene expression analysis among multiple tissues and, for this reason, was selected as standard control. For example, 36B4 has been used for normalization of RT-qPCR data analyzing gene expression changes during adipogenesis (Ramlee et al., Cell Cycle, 2014), as well as during osteogenesis (Darini et al., Oncogene, 2012).

Point 4: Figure 5: Please explain where is the control group? Is it normalized to all the groups? What is the relative cell growth? Why from 1 to 7 days there is no relative cell growth and then suddenly at 14 days there is a great increase? Maybe the normalization is not efficient. For MTT 20000cells/cm2 is a high cell starting number, depending on the well size. More details on the experiment are required.

Response 4: In order to determine whether surface topography affected cell growth, we performed MTT assays. These colorimetric assays analyze the number of viable cells cultured under different conditions. The measure absorbance directly correlates to the number of viable MSC cells and the values obtained from cells cultured on FLAT PLLA at day 1 were considered as a reference and set at 1. The relative cell growth of each condition was normalized with respect to this reference point and it is displayed as fold change. Sections 2.3 (line 133) and 3.2 (line 251), the Y axis label and the caption of this figure have been rephrased to clarify this point:

“Figure 5 shows the relative cell viability on these surfaces as a function of the culture time, measured as the number of cells per area (cells/cm2) normalized compared to cell growth on FLAT PLLA at 1 day of incubation.”

“Figure 5. MSC proliferation (cells/cm2), as quantified by MTT assays, on 4 types of surfaces and at different incubation times. Cell growth was normalized to the absorbance obtained with cells cultured on FLAT PLLA after 1 day of incubation, which was set as 1. The types of surfaces included: PLLA treated by laser to increase the surface roughness (ROUGH PLLA), and PLLA patterned by grooves of width = 10 µm, depth = 4 µm and spacing = 15 µm (GROOVES 1) and 25 µm (GROOVES 2) compared to FLAT PLLA (b).”

Regarding MSC cell growth kinetics, it was observed that growth was modest during the first week and then increased after 14 days. This is similar to other reported findings, for example, Prosecká and colleagues cultured MSC cells on PCL/PVA nanofibers and displayed a similar growth pattern, with relatively slow growth during the first week. In this case, they analyzed the effects of HA deposition, which caused a marked increase in cell viability, nevertheless, in each condition, growth at day 7 was similar to day 1, and only at day 14 significant cell growth was observed (Prosecká et al., J Biomed Biotechnol, 2012). This information has been added in the Suplementary material, in the description paragraph of figure S1.

Point 5: Figure 7: Higher magnification images would show a clearer focal adhesion view related to vinculin.

Response 5: This is a good suggestion from the reviewer, which we already attempted. However, due to the set-up of our microscope, it was not possible to obtain higher magnification images because surface topography modifications and material did not allow proper focus of the microscope.

Point 6: Section 3.3: MSCs cultured only with adipogenic induction medium do they follow the same pattern?

Response 6: We did not analyze the effect of surface topography on cell morphology considering the adipogenic induction medium alone as culture medium, so we cannot assure that the response of the adipocytes to the topography will be the same as that observed on mix medium, but we did observe that lipid vacuoles in adipocytes were very sensitive to topological edges considering different types of topographies, so we could assume that this behavior is characteristic of the cell type and the surface topography and will not depend on the differentiation induction conditions.

Point 7: Figure 9: Images from control groups would be a useful addition in this figure (MSCs with adipogenic induction medium only and MSCs with osteogenic induction medium only but on different substrates)

Response 7: We did not analyze the effect of surface topography on cell morphology considering the adipogenic or the osteogenic induction medium alone as culture medium.

Point 8: Figure 11:   It is seems that in the mix medium and on grooves you have more osteoblasts than adipocytes. In those cases how do you do the normalization with the endogenous control (36B4 gene) since you cannot distinguish between 36B4 mRNA form osteoblasts and 36B4 mRNA from adipocytes?     

Response 8: The expression levels of the endogenous control gene, 36B4, were comparable, independently of the type of induction media used (normal, adipogenic, osteogenic, or mixed) or the topography of the PLLA. Therefore, normalization with 36B4 was done in all cases against the expression levels of cells grown on FLAT PLLA surface and cultured with normal growth media. 

Point 9: Minor Corrections

Line 54:  “… The last years have witnessed…”

Line 323: “A modest, although not statistically significant…”

Line 341: Change phrasing to better define the composition of the mixed medium.

Line 436: “…, particularly on grooves, in respect to…”

Line 449: “… but none or little in width compared to the flat surface.”

Line 459: “… at the same extent as the pure osteogenesis….

Response 9: All the suggested minor corrections have been included in the manuscript.

Reviewer 3 Report

The authors are describing an interesting and novel approach for surface improvement of PLLA polymer in order to induce cells anchorage, differentiation and guided cell organization. Overall, the paper followed a complete, detailed, rational and well-written course, along with wide complementary testing methods, that resulted in an actually qualitative research study. Therefore, minor aspects are to be revised, as following:

Please provide the authors name abbreviations in the Affiliation section to correlate with the Authors Contributions one.

According to `Authors Guide`:

Graphical abstract is not mandatory, but it could increase the article value and provide an overview image of the Materials and Methods & Results sections.

Abstract: please revise this section following the 3 subsections required.

References: please provide the DOI identification for references 3, 6, 29 and 41.

For all measurement units, along the article, please modify the numbers as superscripts (e.g. m2 → m2).

On section 2.2 please provide, besides the instruments models, the location/country for SEM, FIB, XPS and FT-IR.

Author Response

Point 1: Please provide the authors name abbreviations in the Affiliation section to correlate with the Authors Contributions one.

Response 1: Authors name abbreviations have been included in the affiliation section.

Point 2: Graphical abstract is not mandatory, but it could increase the article value and provide an overview image of the Materials and Methods & Results sections.

Response 2: We have submitted a graphical abstract.

 Point 3: Abstract: please revise this section following the 3 subsections required.

 Response 3: The abstract has been modified according to the instructions for authors, including the following sentences related to the background of the study.

“New strategies in regenerative medicine include the implantation of stem cells cultured in bio-resorbable polymeric scaffolds to restore the tissue function and be absorbed by the body after wound healing. This requires the development of appropriate micro-technologies for manufacturing of functional scaffolds with controlled surface properties to induce a specific cell behavior.”

 Point 4: References: please provide the DOI identification for references 3, 6, 29 and 41.

 Response 4: The DOI has been provided for the mentioned references as suggested.

Point 5: For all measurement units, along the article, please modify the numbers as superscripts (e.g. m2 → m2).

Response 5: Done.

Point 6: On section 2.2 please provide, besides the instruments models, the location/country for SEM, FIB, XPS and FT-IR.

 Response 6: The information required has been added.
